# Strategies to Develop the Use of 4R Intermodality as a Combination of Rail Motorways and Motorways of the Sea

**Alberto Camarero Orive** [1,*] , **José Ignacio Parra Santiago** [2] , **David Díaz Gutiérrez** [2] **and Francisco De Manuel López** [1]

1 Departamento de Ingeniería del Transporte, Territorio y Urbanismo, ETSICCP, Universidad Politécnica de Madrid, 28040 Madrid, Spain; francisco.demanuel@upm.es
2 Departamento de Arquitectura, Construcción y Sistemas Oceánicos y Navales, ETSIN, Universidad Politécnica de Madrid, 28040 Madrid, Spain; joseignacio.parra.santiago@alumnos.upm.es (J.I.P.S.); david.diaz@upm.es (D.D.G.)
* Correspondence: alberto.camarero@upm.es

**Abstract:** This paper introduces the concept of R4 (road-rail-ro-ro), a concept increasingly used in transport and logistics research circles that defines the modern concept of the transport chain as it passes through the intermodal use of rail, road, and ship via ro-ro. The integration of the new rail-road freight services into the reference supply model allows us to define the supply model for the design scenario on which the evaluation is now focused in terms of service-mode demand shares and in terms of design network flows and performance indicators carried out by demand-supply interaction models applied to all available service-modes. The use of strength-weaknesses-opportunities-threats analysis (SWOT) allows for the identification of some strategies to enhance and improve the current rail and maritime corridors in order to attract more customers using the different services, ultimately triggering the involvement of more actors in generating bigger and better integrated logistic chains using intermodality. The SWOT analysis allows the identification of a series of measures in order to adapt, maintain, enhance or exploit the aspects arising from the expert analysis.

**Keywords:** intermodality; combined transport; rail motorways; ro-ro; rail-road; motorways of the sea

## 1. Introduction

This paper presents the concept of R4 (road-rail, ro-ro), a concept increasingly used in transport and logistics research circles that defines the modern concept of the transport chain, through the use of rail intermodality, the use of the last mile (road) and sea transport (ship via ro-ro).

The aim of this research is to define how to act to promote this type of intermodality (R4). To do so, a simple but effective tool has been used, namely a classic SWOT analysis that helps us to see the key aspects of this intermodality and to be able to develop strategies in its first steps as a new concept on the rise in Europe.

Transportation has always played a relevant role in all times and societies, according to its means and possibilities, although nowadays it has a special relevance due to the globalization process of the world economy, with a growing economic competitiveness that demands a special effort from all countries. In addition to its economic relevance, it is also an important factor for social and territorial cohesion. The European Union (EU) is aware of the importance of transport and the need to address it in a joint and comprehensive manner throughout the EU, avoiding local or national approaches, and has therefore opted to create a transport area at European level to maintain the competitiveness of this area within world trade.

In order to boost the transport of goods and passengers between the different European countries, the European Union has proposed corridors that provide the backbone of transportation activities across the territory. These rail corridors also have an intermodal

conception, so that once they are in operation they will connect with the most important ports, airports and roads, linking most of the nerve centers of European transport.

In recent decades, changes in scale on the sea side of transport (e.g., with the increase in size of container ships and large global shipping alliances) have not been matched on the land side, making it increasingly important and differentiating to optimize transport chains on the land side, both in terms of time and cost. Currently, a vertical integration of shipping lines is taking place and they are beginning to be present throughout the entire transport logistics chain, especially in the rail mode of transportation, in addition to the maritime mode of transportation.

One of the main objectives of ports, in addition to improving their connectivity with other world ports, is to expand and optimize accessibility to their land area of influence or hinterland in order to achieve a greater uptake of import and export traffic. In this sense, a rail can become a key tool for ports to improve their share of semi-trailer traffic in their hinterland, seeking to extend maritime ro-ro traffic services to the hinterland and freeing the carrier from long distances of approach or dispersion to/from the port [1].

It is true that rail is more competitive for the transport of containers to/from the hinterland than semi-trailers, but there are still important opportunities to take advantage of the benefits of rail transport also for the transport of semi-trailers to/from the hinterland of European ports [2] and, potentially, of Spanish ports. In the case of Barcelona and Algeciras, rail traffic combining semi-trailers with rail is being promoted and, as port nodes, they also favour even more access to ro-ro traffic, which is the aim of this article with the defined R4.

### 1.1. Case of Spain

In the case of Spanish ports of general interest, according to the statistics of Puertos del Estado, containerized import and export traffic in 2021 totaled 6,281,898 TEUs, with the ports of Valencia (with 2.5 million TEUs), Barcelona (with 1.8 million TEUs) or Bahía de Algeciras (0.7 million TEUs) being the most significant with more than 79% of the market.

Road freight transport has a clear preponderance in the internal movement of goods by land modes. In particular, according to data from the Permanent Survey of Road Freight Transport (EPTMC), road transport reached a modal share of 95.0% in 2019 in terms of ton-kilometers, with the weight of the rail mode being barely 5.0% as can be seen in the following graph.

In 2021, international short sea shipping (SSS) of ro-ro cargo (excluding goods vehicles) regained the dynamism of previous years, after the reduction (3.2%) of 2020 due to the COVID-19 pandemic crisis. The increase has been very pronounced (25.7%) compared to 2020, but also compared to 2019 (21.6%) consolidating the recovery. However, the behavior has been different between facades: the Atlantic facade increased by 18.1%, while the Mediterranean recorded an increase of 26.9%.

On the Atlantic façade, the growth in traffic with Ireland—which has almost doubled (93.4%) and also with Belgium (11.0%)—stands out very significantly. In both cases the figures are higher than in 2019. In the case of the United Kingdom, the increase recorded (14.2%) allows almost matching the 2019 figure, although not in the case of France (16.1%).

On the Mediterranean side, the growth in traffic with Morocco (30.1%) and Italy (20.5%) stands out, while with Algeria and Tunisia traffic has decreased by 36.1% and 3.5%, respectively.

### 1.2. Problems of Intermodal Transport in Spain

The current panorama of intermodal transport in Spain is mainly characterized by the following features:

○ Low share of total freight transport in Spain, which contrasts with the general theoretical agreement on its analytical advantages over other modes of transport.
○ The existence of problems arising from the inadequacies and rigidities inherent to conventional rail operations in Spain, which increase operational costs and limit management improvements, such as:

- The length of the trains, which in most cases is 450 m in Spain compared to 700 m in other European countries and the change of gauge at the Spanish-French border.
- The terminals are insufficient and poorly located in relation to the main freight generation/attraction points and lack the necessary characteristics and features.
- Priority is given to passenger trains, which is particularly burdensome in the major freight generation/attraction centers.
- Institutional and regulatory schemes make it difficult to rationalize the dimensioning and use of production resources.

There is currently significant and growing competition between the different operators, both public and private, without this competition having translated into increases in rail and intermodal transport market share. All of this is taking place in a global scenario where some elements and bases for foresight appear to be partly contradictory:

- The sustained development of container traffic and the configuration of a dense and connected network of this type of transport on a global scale.
- The growing congestion of road networks (Spain's connections with France and central Europe), with accelerated growth in the external costs of road transport and an increasing awareness of its impossible sustainability in the medium and long term.

## 2. State of Art

Intermodality is a very important operation in transport management [3], especially in most cases of automotive supply chains. There are several typical examples of intermodal transport [4] and one of them is known as "ro-ro" (roll-on/roll-off) since trucks or cars roll on and off ships. This paper considers ro-ro intermodality, which is evident in the port movements of cars and new vehicles [5], and its relation to the outbound inventory of the automotive supply chain. In special port terminals, cars are driven to or from special vessels called car carriers [6]. Car-carriers, according to [7] can be described as "Ro-ro vessels are something similar to large floating car parks. They have large doors on their sterns or sides. Ramps are used for transport and are extended to the shore". Fischer and Gehring [8] describe in detail the planning of vehicle transshipment at a car terminal in a seaport using a multi-agent system. In addition, car carriers consist of four, five, six or more parking areas with a capacity of up to eight thousand cars or more. Roll-on/roll-off vessels have an important advantage: flexibility. However, they require a demanding stowage operation to load cars to or from the decks [9]. The geographical definitions found for SSS cover ports located along the coastal zone of European Union (EU) and non-EU countries (including islands), from the Barents Sea and Scandinavia, through the North Sea and the Baltic Sea, to the Mediterranean, including also some ports in North Africa and the Black Sea [10]. Therefore, it makes sense to associate SSS with the maritime transport of cars to European ports, linking focal car production with consumer markets [11].

The transport policy reference framework consists of the transport interventions contained in planning and programming documents that can reasonably be assumed to be completed in the long-term time horizon (e.g., 2020). It is the basis of the reference offer model through which the level of service (LoS) attributes are calculated for each alternative service-mode. The LoS attributes and new rail terminals contribute to identify potential rail traffic corridors which mainly refer to origin-destination (O-D) pairs characterized by the presence of rail terminals and by a relevant future freight demand served by road, as rail-road transport is not as attractive due to lack of services.

The integration of the new rail-road freight services into the reference supply model allows us to define the supply model for the design scenario on which the evaluation is now focused in terms of service-mode demand shares and in terms of design network flows and performance indicators carried out by demand-supply interaction models applied to all available service-modes. In the following, for the convenience of the reader, we will use the generic terms rail and maritime to indicate combined rail-road and combined maritime Ro-Ro transport, respectively [12].

### 3. Transport Challenges Involved in the Use of R4

From the different studies and documents consulted, it can be deduced that the main advantage of intermodal transport is the possibility of combining the advantages inherent to the different modes of transport involved. Of the factors in favour of intermodal transport, the cost factor stands out above the rest. The economic effects of intermodal transport to be highlighted can be grouped into two blocks:

- Reduction of social costs: road safety, air pollution, noise pollution, energy and raw material consumption.
- Reduction of infrastructural costs: reduction of road traffic, with the consequent reduction of congestion, and better use of the current capacities of transport systems.

Other factors can also be highlighted, such as the ability to carry large volumes of freight over long distances, as well as the possibility of carrying out maritime or rail transport during weekends, holidays or nights. However, in order to make intermodal transport a real alternative to unimodal road transport, the friction costs of switching modes must be identified, quantified and reduced. These are extra costs that are a measure of the inefficiency of intermodal transport operations and translate into higher prices, longer delays and less reliable delivery times, less availability of quality services, limitations on the type of goods, more risks of damage, more complex administrative procedures, etc.

The promotion of intermodality is a basic instrument for achieving a traffic reduction in the Spanish road system. This implies a reorganization of resources in companies, in which the focus would become complete the transport chain and the lorry would occupy the collection and delivery links. This strategic reorientation of the current unimodal road transport operators would, in turn, be an important contribution to the development of intermodality, together with improvements in the operation of rail and maritime modes.

At the same time, the actors involved in the transport chain must be able to provide added value to the chain itself to also help eliminate friction costs. This process includes services such as warehousing, information management, etc. Other needs include a need to standardize the different loading units of intermodal transport and to unify responsibilities for intermodal transport between countries.

In another study carried out by Buck Consultants International, Deloitte and Touche Bakkenist [13], the following four decision-making levels are identified according to the ability to manage the functions carried out in the transport chain.

(a) Flow of goods: the company determines the quantity, type of goods, frequency of shipments and quality required. This function is usually carried out by the charterer.
(b) Transport flows: the company must comply with the characteristics demanded of it, such as origin, destination and types of goods, etc. These companies must design the optimum logistical procedure appropriate to their characteristics. This type of function is usually carried out by the carriers.
(c) Responsible for transport planning: the company assigns both shipments and travel plans to the load units. This function is usually also performed by the freight forwarder.
(d) Transport: the company assigns a transport unit and a driver and does not have any organizational capacity.

### 4. Methodology

The methodology used in this paper is based on a panel of experts brainstorming ideas to help identify the key aspects of the SWOT. The analysis of the SWOT exercise results will highlight those factors that are most repeated throughout the experts' contributions, based on aspects to be taken into account from the point of view of Spanish intermodality, not extrapolated to all European intermodality.

### 4.1. Expert Panel

In the expert group, prior to the SWOT analysis, issues that may affect the development of intermodality and the R4 concept have been discussed. These experts are recognized within the sector to discuss the different aspects that can be included in the SWOT matrix.

The experts are from private companies related to the maritime sector, from the public company in charge of the management of the Spanish ports, from prestigious research and academic staff, and from independent expert assistants and consultants (see Figure 1).

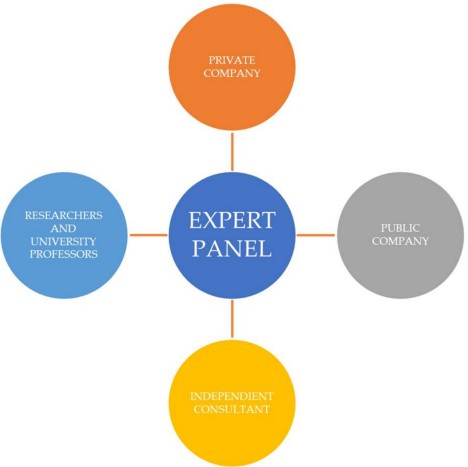

**Figure 1.** Expert panel composition. Source: Own elaboration.

The methodology used to gather the contributions of panel of experts was as described next. A total of 28 people (made up of private companies in the sector, universities and research centers, independent consultants and public companies) were given a survey with different factors for them to develop the key aspects (strengths and weaknesses, as well as improvements or other comments) to bring to their attention. Once these factors were all gathered, those that had come up in the largest number of surveys were used in a second round and they panel experts were asked to score them in order to identify those to be considered in our SWOT.

The panel of experts is always a key element in any research development as they contribute ideas and visions from various points of view/fields, which further enriches our research.

### 4.2. SWOT Analysis

The following figure (Figure 2) represents one of the tools used in the methodology of this article, the SWOT (strengths, weaknesses, opportunities, threats) analysis.



**Figure 2.** SWOT Panel. Source: Own elaboration.

The opportunities and threats correspond to factors external to the organization while the strengths and weaknesses belong to the internal sphere. The correct identification of these factors allows the construction of anticipated scenarios to rectify deviations from the company's objectives.

The working scenario where the analysis is applied is determined by means of the SWOT matrix. That scenario is the current concept of R4 and this work aims to study the possibility of the concept being exploited to its full potential by means of research with real data, which allows proposals to be developed to facilitate port planning and strategies to be formulated [14].

As far as the sources chosen are concerned, they have been based entirely on official websites. Thanks to them, it has been possible to capture all the necessary aspects in order to achieve the approach that the study pursues. The use of this tool makes it possible to establish the current scenario on which to act in order to achieve a correct use of the R4 concept.

For this paper, the SWOT Methodology of the Ministry of Industry, Trade, and Tourism (MICT) of the Spanish Government has been used, which allows access to a technological tool that favours the creation and management of the SWOT matrix to be generated.

The first phase consists of the study of the concept of intermodality and what its definition encompasses, which makes it possible to extract information that serves to see how and which actors in the transport chain can be affected (in our case, road-train and ro-ro).

The second phase consists of the SWOT analysis. A classic analytical methodology was used to collect and analyze the data. This first part is based on extracting information from the data. To do so, an analysis of the results obtained from the SWOT panel is carried out [15].

Once the four elements have been identified, the SWOT impact matrix is drawn up to evaluate the intensity of interaction between the external and internal elements. To do this, a numerical value is assigned proportional to the intensity of the impact on the interception of the coordinates that identify each element. The quadrant with the highest score defines the situation in which the company is appreciated and the sums by axes identify the real impact of each element.

Depending on the quadrant with the highest score, the matrix identifies four conceptually distinct alternatives for the definition of the strategy which, in practice, may overlap:

- The objective of the WT (weaknesses vs. threats) strategy is to minimize weaknesses as well as threats. An organization located in this quadrant would be facing its worst situation regarding the achievement of its objectives, its main efforts would have to be devoted to fight for its survival or it would irremissibly reach its definitive liquidation. As alternative strategies, one can assume the reduction of operations in order to minimize weaknesses or wait for changes in the environment that make threats disappear, the latter at a high risk of not being successful. Whichever strategy is selected, the DA position will be the most dangerous and it is suggested to assume a survivalist attitude.
- The second strategy, WO (weaknesses vs. opportunities), requires minimizing weaknesses and maximizing opportunities. A company in this situation identifies the opportunities offered by the environment, but recognizes that its organizational weaknesses do not allow it to take advantage of them. A strategy variant may be to miss the opportunity that the competition will most likely take advantage of. In this situation the company must assume an adaptive position.
- The ST strategy (strengths vs. threats) is based on the assumption that the institution's strengths can cope with the threats in the environment. Its objective is to maximize the former while minimizing the latter. This situation does not necessarily mean that the organization has to look for threats in order to deal with them. On the contrary, the company's strengths should be applied at its discretion and in a timely manner. The correct position for a company in such a situation is defensive.
- The SO (strengths vs. opportunities) situation constitutes the most advantageous quadrant, where all companies would like to be positioned to use their strengths to exploit opportunities. This success-oriented situation suggests taking an offensive stance.

The SWOT analysis evaluates a specific situation conditioned to the external and internal elements that coincide at a certain moment in the life of the organization. The constant changes of the evaluated elements induce the need to periodically perform the SWOT analysis in order to adjust the strategic formulation to respond to the new situation of the environment and the organization itself [16].

SWOT analysis is currently used in all fields [17] where decision-making is an important aspect, such as environmental sustainability [18], adoption of Industry 4.0 [19], quality management in higher education [20] and risk assessment [21]. However, the common problem is that the criteria factors cannot be measured quantitatively, which makes it difficult to determine which variable mainly influences the strategic decision [22].

The SWOT matrix provides an overview of the internal and external situation of a company. The results help to know the capabilities and identify the challenges (present and future) to be addressed in a company [23] (see Figure 3).

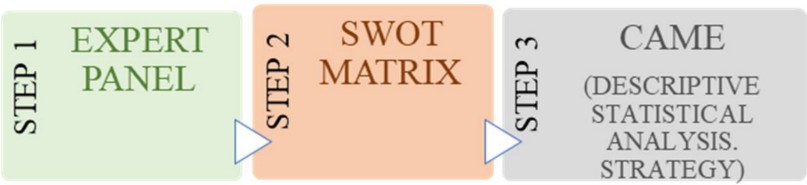

**Figure 3.** Methodological process. Source: Own elaboration.

In short, the SWOT analysis is able to highlight the different situations that any company incurs and anticipates the different possible scenarios. It allows you to develop a strategy that ensures the fulfilment of the objectives set [24], thus turning the SWOT matrix into an effective tool that facilitates the appreciation of the environment and the improvement in decision-making [16].

## 5. Swot Results and Strategies

The results of this study, as mentioned above, have been drawn from a meticulous analysis of the questionnaire answers of experts in the field of intermodality, ro-ro traffic and short-distance freight transport, as well as researchers and staff of publicly owned companies in the logistics-port sector.

For this purpose, the following SWOT matrix (Table 1) was obtained using the SWOT tool of the Spanish Ministry of Industry, Trade, and Tourism (MICT).

**Table 1.** SWOT Results. Source: Own elaboration.

| Strengths | Weaknesses |
|---|---|
| Modal integration of transport | Costs of using various forms of transport |
| Use of the semi-trailer or full truck as a freight transport unit | Existence of few infrastructures adapted to railways |
| Support for european initiatives | Lack of general regulations |
| **Threats** | **Opportunities** |
| Efficiency of other transport modes | Current growing demand |
| Lack of infrastructure for r4 development | New models of business collaboration |
| Cost generated by inefficiencies | Reduction of carbon footprint |

The results obtained (Table 1) tend to define a problem associated more with the costs of using various forms of transport, associated with "just in time" trade or related to the so-called "door to door", which often means that the transport chosen is solely and exclusively by lorry, which is transported by road in the majority of cases. This is due to the associated costs that cannot be passed on to the end customer (or must not be passed on in order to keep the customer) and to fixed times, i.e., the timetables of certain ro-ro routes or the "rolling road" (if we are talking about rail). In other words, there may sometimes be delays in the time it

takes for the goods to reach their destination and be delivered. These are costs that are borne by the transport company, without being passed on to the end customer.

It is worth noting that some of the aspects considered in the SWOT coincide with the challenges that some research and authors have identified, as discussed in the sections above. This indicates that the trend towards R4 intermodality continues and that we are on the right track in developing this logistics cooperation mechanism between different transport actors.

Finally, the tool used by the Ministry allows us to relate aspects of the SWOT and to be able to generate or extract strategies (Table 2) that should be taken to continue promoting R4 as an intermodal transport method in Spain and in Europe.

**Table 2.** Strategies. Source: Own elaboration.

| Strategies | |
|---|---|
| ADAPTATIVE | Adapting/adopting european intermodality guidelines |
| DEFENSIVE | Maintaining and promoting european corridors |
| OFFENSIVE | Attraction and creation of measures to promote 4r in european and national transport |
| SURVIVAL | Optimisation/creation of intermodal 4r routes |

Some of these strategies are to strengthen and improve the current rail and maritime corridors, which will make it possible to attract a greater number of customers who use the different services. This will lead to more agents involved in generating bigger and better integrated logistics chains in the use of R4 intermodality.

Table 2 describes the different strategies to be followed, supported by some of the aspects highlighted in the previous SWOT. Broadly speaking, it is based on the aspects or guidelines that Europe can provide in terms of intermodality. The European Union is currently promoting this type of transport, which allows the article to consolidate this current trend.

## 6. Conclusions

A methodology based on a questionnaire used to carry out a SWOT analysis of R4 intermodality has been presented and used to meet the research needs of being able to evaluate this concept. The general conclusion is that the potential for the development of intermodality is high, as it still has a long way to go to become the main part of the logistics chain. Large shipping companies are betting on rail as a second means of transport to distribute their goods in origin and destination nodes.

Promoting the development of regular ro-ro lines (i.e., motorways of the sea) is a still a main priority. This line of action aims to promote, among road transport companies, the use of vessels and services designed to transport lorries and semi-trailers directly by ship. This action converts maritime transport into an extension of road transport, ensuring that part of the journey made by the lorry is made by ship, thereby helping to reduce the emissions associated with road congestion, particularly at the border crossings between Spain and France.

Promotion of rail transport to and from ports is also an important line of action. It seeks to take advantage of the greater environmental efficiency of rail traffic compared to road traffic, enabling and promoting the use of rail for those flows of goods originating from or arriving at ports which, either due to their concentration in large volumes, or since they cover long distances with sufficient concentration of freight at origin/destination points, are appropriate for this transport scheme.

European guidelines, European corridors and the creation of new intermodal routes will promote the R4 concept in countries where maritime transport has been developed and promoted together with rail. In addition, current motorways of the sea routes will be

affected and/or promoted towards this R4 concept, linking ro-ro with rail for the transport of cargo by lorry along trans-European rail freight transport routes.

**Author Contributions:** All authors have contributed equally to the development of this paper, contributing ideas, reviewing progress and collaborating in the writing of the paper (all sections). All authors have read and agreed to the published version of the manuscript.

**Funding:** This research received no external funding.

**Institutional Review Board Statement:** Not applicable.

**Informed Consent Statement:** Not applicable.

**Data Availability Statement:** Not report any data.

**Conflicts of Interest:** The authors declare no conflict of interest.

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
