# Peer review of "Strategies to Develop the Use of 4R Intermodality as a Combination of Rail Motorways and Motorways of the Sea"

_jmse, doi:10.3390/jmse10070972_

Round 1
Reviewer 1 Report
The Authors present a contemporary problem, i.e. the strategies to develop the use of 4R intermodality. The proposed methodology is a combination of experts’ panel and SWOT analysis. My greatest concern is the lack of novelty. Moreover, the considerations only touch the problem without going into details.
I would like to raise the following issues:
1. Highways (rail and of the sea) are only presented in the title and keywords.
2. The abstract should clearly state the novelty of presented approach and the result, which has been achieved.
3. Some parts of the manuscript are ambiguous or not clear, e.g.:
- Comparison of the fields’ location in figure 2 and table 1 within the SWOT analysis is different. I assume that their location is important while considering the results of analysis in the fields’ combination context. This aspect is omitted during the practical example presentation, as well.
- SWOT results are very limited (see table 1). There are presented only 3 conclusions per SWOT field arising from 28 experts.
- It is not clearly stated, which part of the manuscript reflects Authors’ own methodology and research.
- The conclusions concentrate mostly on promotion of R4, while the results of SWOT analysis and strategies presented in table 2 present the other aspects, as well.
4. There are a lot of references missed, e.g. the sentences in lines 33-40.
5. Information presented in the following lines is not clear: 202-204, 325-329
6. The sentence in lines 161-162 is not finished.
7. After the beginning of the sentences in lines 91 and 111, the bullets are not distinguished in the next lines. Therefore, it is not possible to recognize the end of those sentences.
8. There is no reference to figures in the text.
9. Most of the figures are oversized.
10. Information presented in figure 1 is in Spanish (not English).
I have the questions such as:
1. What is “high of the sea” (see keywords) in the context of the manuscript?
2. What is the object of SWOT analysis (see line 15)?
3. The Authors mention in the conclusions that “the methodology used has been able to meet the research needs of being able to evaluate the current concept of R4 intermodality”. What is the justification?
4. Is the promotion of R4 the best way to convince transport companies or operators to change their way of work?
5. What are the further steps of the research, which are usually presented in conclusions?
The paper is well written. However, English changes, including punctuation, spelling and grammar are required as well as some editing corrections.
In my opinion, a profound revision of the manuscript should be carried out.
Author Response
We have responded to all the points raised, certain paragraphs of the article have been modified to respond to all the reviewers and their comments to make this paper better.
Round 2
Reviewer 2 Report
The authors improved the paper. Now I have no comments.
Author Response
Thank you that you agreed to accept our paper.